# Count Me in, Count Me out: Regulation of the Tooth Number via Three Directional Developmental Patterns

**DOI:** 10.3390/ijms242015061

**Published:** 2023-10-11

**Authors:** Zheng Fang, Devi Atukorallaya

**Affiliations:** Department of Oral Biology, Dr. Gerald Niznick College of Dentistry, Rady Faculty of Health Sciences, University of Manitoba, Winnipeg, MB R3E0W2, Canada; fangz6@myumanitoba.ca

**Keywords:** tooth number, tooth agenesis, supernumerary tooth, deciduous tooth, permanent tooth, diphyodonts, dental lamina, continual lamina, successional lamina

## Abstract

Tooth number anomalies, including hyperdontia and hypodontia, are common congenital dental problems in the dental clinic. The precise number of teeth in a dentition is essential for proper speech, mastication, and aesthetics. Teeth are ectodermal organs that develop from the interaction of a thickened epithelium (dental placode) with the neural-crest-derived ectomesenchyme. There is extensive histological, molecular, and genetic evidence regarding how the tooth number is regulated in this serial process, but there is currently no universal classification for tooth number abnormalities. In this review, we propose a novel regulatory network for the tooth number based on the inherent dentition formation process. This network includes three intuitive directions: the development of a single tooth, the formation of a single dentition with elongation of the continual lamina, and tooth replacement with the development of the successional lamina. This article summarizes recent reports on early tooth development and provides an analytical framework to classify future relevant experiments.

## 1. Introduction

Tooth number anomalies are a common developmental malformation. In humans, the prevalence of supernumerary teeth—also known as hyperdontia, the development of an excessive number of teeth in the oral cavity—is 0.1–3.8%. This condition is more prevalent in males than females and in the permanent dentition than the deciduous dentition [1,2,3,4]. Supernumerary teeth are more common in the anterior maxilla [5]. Tooth agenesis refers to the absence of one or more teeth during development [6]. In humans, the prevalence of tooth agenesis is about 2–9%; it is more common in females [7,8]. In the clinic, tooth agenesis is classified as hypodontia (less than six teeth missing), oligodontia (no less than six teeth missing), and anodontia (absence of the entire dentition). The third molar teeth are not considered for this classification because they are the most commonly missing teeth [6]. The mechanism of tooth number alterations in early odontogenesis is an important but still unresolved puzzle in dentistry as well as developmental biology.

Contrary to its straightforward implication, tooth number regulation is a complicated topic in the field of dentistry. First, numerous signalling pathways are involved in the early development of teeth [9,10,11,12,13]. These pathways have diverse roles that have not been fully described. Second, teeth and dentitions vary among species, and it is intuitively unreliable to extrapolate results from animal models such as mice to humans, as they are different in dentition patterns inherently. In addition, the number of teeth is altered during dentition replacement. In humans, the deciduous dentition comprises 20 teeth while the permanent dentition includes 32 teeth. Another diphyodont example is the ferret (*Mustela furo*), which has 28–30 deciduous teeth and 34 permanent teeth [14]. Polyphyodonts such as teleost fish replace their teeth throughout their entire lives. In the Mexican tetra (*Astyanax mexicanus*), the oral tooth number increases within 30 days post-fertilization and becomes constant [15]. Obviously, researchers have investigated many regulatory mechanisms in early tooth development, but much less is known about tooth number regulation, especially when considering the tooth development patterns from a three-dimensional view [16,17,18].

In this review, we summarize the findings of the tooth number regulatory network in early tooth development and propose three directions for tooth number regulation, considering the general array of teeth in the oral cavity (Figure 1). The first direction explains single-tooth development. Generally, it involves the process from the initiation of oral epithelium thickening to the late developmental stage of the dental placode, the stage at which there is an increase in the number of teeth. Arrest at this stage would decrease the number of teeth. The second direction outlines the development of several teeth within one dentition along the alveolar bone: The distal extension of the dental lamina gives rise to more teeth, and when this process stops, no more teeth are formed. In the third direction, we discuss how replacing a functional tooth with a successional one has a potential effect on the consistency or inconsistency of the tooth number. This review provides a novel categorization of tooth number regulation from a three-dimensional perspective and should help researchers reassess the previous relevant work.

## 2. The First Direction: Initiation of Odontogenesis in the Tooth-Bearing Bones

In the development of ectodermal organs such as the hair, glands, and teeth, there are reciprocal interactions between the epithelium and the underlying neural crest–derived ectomesenchyme [19]. Tooth development has been well conserved during vertebrate evolution [20]. Briefly, during early embryogenesis, cranial neural crest cells migrate from the dorsal epithelium of the neural tube into the pharyngeal arches and form the jaw structures. Then, the overlying epithelial–mesenchymal interactions lead to the formation of dental placodes, which initiates odontogenesis [21]. The placodes undergo sequential morphological changes that include the bud, cap stage, and bell stages. The process during the bell stage and biomineralization determines the morphological characteristics of the presumptive tooth jointly [22,23]. A tooth germ develops into a functional tooth, meaning that the tooth number increases by one, and failure of tooth formation results in a decrease in the tooth number. The steps from the presumptive tooth germ present as a thickening epithelium to the cap stage are considered early tooth development and are also regarded as the first direction in tooth number regulation. Obviously, all teeth—anterior and posterior, deciduous and permanent, oral and even pharyngeal—undergo similar later developmental processes, including the bell stage, maturation, eruption, etc. Hence, these steps could be considered an extension of the first direction of the developmental network (the “broad” first direction). In this review, however, we consider that the first direction applies only to those teeth that develop from only thickening the oral epithelium in the deciduous dentition, unless stated otherwise.

### 2.1. Tooth Number Regulation at the Early Dental Placode Formation Level

The formation of the primary epithelial band is commonly considered to be the initiation of tooth development. In the mouse, cranial neural crest cells start to migrate to their destination at embryonic day 8 (E8) and complete this migration by E9.5 [24]. Paired-like homeodomain transcription factor 2 (PITX2) is the earliest transcription factor expressed during tooth development [25]. It is expressed in the presumptive primary epithelial band at E8.5 [26]. At E11, a horseshoe-shaped thickening is formed in the oral epithelium. This thickening results from the cellular arrangement of several layers of tall columnar cells with basally placed nuclei [27]. This provides the anatomical landmark for the formation of the presumptive tooth germs [27].

There is the phenomenon that odontogenic genes are expressed along the presumptive alveolar bone and then become condensed in certain areas at the initiation stage in vertebrates. In mouse studies, genes including forkhead box I3 (*Foxi3*) [10], *Pitx2* [11], and SRY-box transcription factor 2 (*Sox2*) [12] are expressed diffusely and strongly over the entire primary epithelial band at E11.5. Then at E12.5, the expression of these genes becomes restricted to the incisor domain and molar primordia: Four discrete areas with intense gene expression in the upper or lower jaw [10,11,12]. Similar patterns have been described in the Mexican tetra. In this species, *pitx2* and sonic hedgehog (*shh*) are expressed uniformly in the oral epithelium at 48 h post-fertilization (hpf), and then intensively expressed in the tooth germ domain at 72 hpf [28,29]. In the Mexican tetra, distal-less homeobox 2b (*dlx2b*) expression marks the oral tooth epithelium at 42 hpf; at 72 hpf, *dlx2b* is localized to two domains, each of which represents a tooth germ [29]. This transformation from diffuse expression to intense localized expression has also been observed in rainbow trout [30]. There is a slightly different pattern in the ferret: Diffuse *PITX2* expression in the incisor region and intense *PITX2* expression in the canine region [20]. The above studies suggest that tooth germ formation tends to initiate at localized areas with upregulated expression of odontogenic genes.

Gene expression precedes the morphological and cellular changes of the primary epithelial band. At E11.5 in murine embryos, those specific areas in the primary epithelial band begin to stratify and evaginate into the underlying dental mesenchyme. At this stage, the dental placode becomes more prominent. Compared with the non-placodal area, epithelial cells in the dental placode have spindles that are mostly oriented perpendicular to the basal layer. Between E11.5 and E12.5, these dental-placodal cells proliferate, which promotes epithelial stratification [31]. From E12.5 on, the dental placode begins to invaginate further into the dental mesenchyme, with central epithelial cells pushed downwards by neighboring cells, independently of cell division [31,32]. The intrinsic factors driving the progression of tooth germ development are not completely known. In 2016, Ahtiainen et al. proposed that the initiation knot, a small population of nonproliferating cells, governs the process [9]. The initiation knot area is more restricted compared with Foxi3 expression but coincides with Shh expression in the dental placode. The authors found a smaller tooth bud when the initiation knot was smaller. These findings suggest that the initiation knot is a signalling centre in early dental placode formation. Subsequently, placodes undergo additional morphological changes from the bud stage until a functional tooth is formed.

### 2.2. Tooth Germ Arrest

Manipulation of specific genes can stop or induce tooth development. Below, we discuss some of the genes important for tooth development.

Bone morphogenetic proteins (BMPs), a member of the transforming growth factor-beta (TGF-β) superfamily, are a group of proteins that exhibit a promotive role in tooth development [33]. Inhibition of BMP signalling arrests tooth development [34]. BMP4 expression changes dynamically during tooth development. Its expression is concentrated in the dental epithelium during the thickening epithelium stage and then shifts to the ectomesenchyme in the dental placode and early bud stage in mice [35]. Transition from the dental lamina stage to the bud stage is interrupted by the overexpression of Noggin, a BMP antagonist [34,35].

There are interactions between BMP and proteins such as Msh homeobox-1 (MSX1) and paired box-9 (PAX9) in the dental mesenchyme. MSX1 plays a significant role in maintaining the odontogenic function of the dental mesenchyme [36,37]. Early studies have demonstrated that deficiency in *Msx1* arrests tooth development at the bud stage in mice [13]. There is a reciprocal loop system established by *Msx1* and *Bmp4* during early tooth development [35]. Early expression of *Msx1* in the dental mesenchyme is induced by BMP4 [38], while later in the dental mesenchyme, *Msx1* induces and sustains BMP4 expression [39]. BMP4 stimulates *Msx1* transcription by inducing the IPO7–Smad1/5 complex that is recruited to the *Msx1* promotor in dental mesenchymal cells [40]. These findings suggest that Msx1 is critical in bud-to-cap formation during tooth embryogenesis. Indeed, in the late stage of tooth development, *Msx1* exerts an inhibitory effect on the expression of *Bmp4*, causing a reduction in odontoblast differentiation, which is different from the role it plays during early tooth development [36].

PAX9 plays a crucial role in the normal development of various organs, including the palate, teeth, and limbs [41]. In mice with *Pax9* mutations, tooth development is arrested at the bud stage, leading to hypodontia and oligodontia [41,42,43,44]. In *Pax9*/*Msx1* double heterozygous mutants, the lower incisor epithelium exhibits a smaller dental papilla, and smaller incisors are formed [45]. Ogawa et al. [46] described a regulatory network involving *Pax9*, *Msx1*, and *Bmp4* in the dental mesenchyme. At E12.5, there is no significant difference in *Msx1* expression between *Pax9* mutant and wild-type mice, while at E13.5, *Pax9* is necessary for the normal expression of *Msx1* and the PAX9–MSX1 protein heterodimer continues to activate *Msx1* and *Bmp4* transcription [46]. An in vitro experiment showed that in the absence of *Pax9*, *Msx1* suppressed *Msx1* and *Bmp4* expression, a finding that suggests the proper protein structure of PAX9 is crucial for tooth germ development [46,47,48].

Wingless-related integration sites (WNTs) are a group of secreted glycoproteins that are involved in a variety of biological processes, including odontogenesis [49]. Dickkopf-1 (DKK1) is an inhibitor of the WNT signalling pathway [50]. Andl et al. [50] generated transgenic mice with ectopic expression of *Dkk1* in basal epidermal cells. They observed arrested tooth development between the epithelium thickening stage and the bud stage. Moreover, *Dkk1* overexpression arrests the development of incisors at the placode stage, while molars are arrested at the lamina stage [51]. WNT signalling is also involved in the bud-to-cap transition downstream of MSX1 signalling [52]. WNT10A and WNT10B are common proteins in tooth development and have been the subject of several studies. A comprehensive review reported that patients diagnosed with bi-allelic *WNT10A* mutations were mostly deficient in teeth of the maxillary dentition: 77% of patients were missing the maxillary permanent lateral incisors and 71% of patients were missing the mandibular second premolar [49]. Although the maxillary central incisors were present in these patients, in a report on patients with odonto-onycho-dermal dysplasia, which also involves bi-allelic *WNT10A* mutations, all permanent teeth were absent, and there was also agenesis of the deciduous teeth [49,53]. These studies indicate that the WNT signalling pathway may play important roles at different stages of tooth development.

Fibroblast growth factor (FGF) signalling regulates odontogenesis from the early stage. At E10.5, *Fgf17* expression indicates presumptive tooth development sites; after that time, a variety of FGF ligands and receptors are expressed in the dental mesenchyme and/or dental epithelium [54,55]. Some of these proteins are vital to tooth formation. In *Fgfr2*-deficient mice, tooth development is arrested at the lamina stage [56]. In *Fgf3*/*Fgf10* heterozygous mutants, molar development is retarded at the lamina stage [57]. Downregulation of *Fgf4* by *Lef* deficiency results in tooth germ arrest [58]. Moreover, FGF signalling regulates early tooth germ development with other signalling pathways, including PAX9 and MSX1 [45,59].

As mentioned previously, SHH is expressed in the dental placode during early tooth germ development. At E12.5, treating tooth germ explants with the Shh inhibitors 5E1 or forskolin and culturing for 3 days disrupted the development of the tooth germ tip compared to the control group [60]. Li et al. [31] proposed that *Shh* expression facilitates invagination of the dental placode in mice. They demonstrated that the treatment of the temporal inhibitor of Shh, cyclopamine, on mouse mandibular explants at E11.5 or E12.5, could prevent invagination of the dental placode without affecting stratification.

Ectodysplasin (EDA), a member of the tumour necrosis factor (TNF) superfamily, is another important factor in initiating tooth development; this signalling pathway occurs upstream of SHH [61,62]. The EDA pathway comprises the ligand (EDA), the transmembrane receptor (EDAR), and the cytosolic signal mediator, the EDAR-associated death domain (EDARADD) protein [63]. EDA signaling may regulate odontogenesis via the EDA/EDAR/nuclear factor-κB (NF-κB) pathway [64]. EDA-A1, an isoform encoded from *EDA*, binds with EDAR, recruits EDARADD and activates downstream NF-κB signaling [65]. The EDA pathway is involved in early tooth development. There are smaller M1 and M2 germs in the *Eda* mutant at E 13, and the overall development of M1 and M2 is also delayed [66]. *Pax9*/*Eda* double heterozygous mutant mice have normal dentition, while *Pax9*+/−/*Eda*−/− mutant mice lose all third molars and mandibular incisors. These findings indicate that *Pax9* and *Eda* have a complementary regulatory network [67]. In dental clinics, the *EDA* mutation is related to both syndromic tooth agenesis and non-syndromic tooth agenesis [65,68]. As *EDA* is localized on the X chromosome, *EDA* mutation causes X-linked congenital conditions, the most common of which is hypohidrotic ectodermal dysplasia (HED); patients with this condition experience severe tooth agenesis, hypotrichosis and hypohidrosis [63,69].

There are several other gene families that play a pivotal role in intact tooth development. For example, *Pitx2* and *Sox2* contribute to the initial knot formation, and deletion of *Pitx2* in the epithelium arrests the tooth germ at the bud stage [25]. Above all, there are many signalling pathways involved in and necessary for normal tooth development.

### 2.3. Diastema in Mice

Disturbance of gene expression in tooth germ development may result in tooth germ arrest. In mice, three molars and one incisor of one dental quadrant are separated, forming a tooth-free zone called the diastema, which researchers agree is caused by tooth germ arrest. The tooth germ has been observed to develop at the early stage in the diastema. At E11.5, the area of the oral epithelial division expands and covers the future diastema gap [31]. Two dental placodes, MS and R2 (anterior and posterior, respectively, in the mandible), of the diastema continue to develop until the bud stage [70,71]. Compared with the normal molar tooth germ, the diastema tooth germ does not exhibit significant morphological differences at E12.5 [72]. However, by E13.5, the diastema tooth germ is notably smaller than the normal dental placode, and there is a lack of mesenchymal condensation [72]. By E14.5, the diastema tooth germ has further reduced in size and appears as a slightly thickened dental epithelium with MS regressed in the mandible and R2 resorbed by the first molar germ in the mandible [72,73].

The potential origins and mechanisms underlying the cessation of tooth development in the diastema have been extensively investigated [70,72]. Recombination experiments have shown that the recombined tissue from the dental epithelium of normal mouse molars and the ectomesenchyme of the diastema fail to give rise to tooth formation [70]. Conversely, when the authors combined the dental epithelium of the diastema with the ectomesenchyme of normal molars, they could rescue tooth formation. In another study, the authors observed that at both E11.5 and E13.5, the dental epithelium and mesenchyme of the diastema tooth germ possess odontogenic competence and the ability to respond to tooth-inducing signals. Moreover, at E13.5, the mesenchyme loses its odontogenic potential (the ability to initiate tooth development) [72].

Taken together, these studies suggest that the mesenchyme plays an inductive role in the diastema. At E13.5, there are differences in tissue-specific gene expression between the diastema and the normal molar dental epithelium. Specifically, *Shh*, *Fgf3*, and *Fgf4*, which are normally expressed in the dental epithelium of the normal molar, are not expressed in the diastema tooth germ [74]. The sprouty gene family, including *Spry2* in the epithelium and *Spry4* in the ectomesenchyme, inhibits the reciprocal signalling between the dental epithelium and the ectomesenchyme, leading to the arrested development of the tooth bud in the diastema [74]. Conversely, suppression of *Spry2* and *Spry4* expression can reverse this phenotype, resulting in normal tooth formation in the diastema gap [74]. Normal production of SHH and FGF4 in the *Spry* mutants might be attributed to the supernumerary teeth in the diastema [27]. Intriguingly, a rescued diastema tooth has also been reported in studies of *Wnt10a*+/−/*Wnt10b*−/− mutants [73] and *Wise* (also known as *Sostdc1* and *Ectodin*)−/− mutants [71,75]. This evidence suggests that the decreased tooth number in the murine diastema is a result of tooth germ arrest at an early developmental stage.

Based on the literature, we consider the phase spanning from epithelial thickening to the cap stage as the first regulatory direction to determine the tooth number. It represents the transition from the absence of one tooth to the initiation of one tooth. This process inherently leads to augmentation of the total tooth number in the oral cavity. The genes we mentioned mostly exert their influence during this developmental period, and any aberrations can impede this process, ultimately manifesting as a reduction in the total tooth number.

## 3. The Second Direction: Progression of Tooth Development in the Mesio-Distal Access

In mammalian oral cavities, teeth are arranged along the alveolar ridge of the maxillary and mandibular jaws, resulting in two dentitions that are formed by the development of a series of individual teeth. The above discussion on the first direction of tooth number regulation describes the concept of the widely accepted ideal model, wherein individual teeth undergo development from epithelial thickening to the cap stage. However, it is important to note that there are some teeth that do not develop as individual tooth germs as in the ideal model. Instead, some teeth develop from a distal extension of the dental lamina, also known as the continual (dental) lamina, that connects to the mesial tooth germ [76]. In 2017, Juuri and Balic [16] introduced the term continual lamina; they defined it as the dental lamina at the horizontal and posterior ends of which teeth are added serially. Before the term continual lamina was introduced, researchers had used the terms posterior lamina, interdental lamina, distal lamina, and additional lamina, among others (Table 1). There has not been a unified consensus on the nomenclature of the continual dental lamina in the literature. In this review, we refer to teeth developing from the continual lamina as continual teeth and their development as continual tooth development. In the following discussion, we consider humans and mice.

### 3.1. Humans

In humans, teeth in the posterior dentition, including premolars and molars, are formed from the posterior progression of the continual dental lamina [76]. The first deciduous molar (dm1) of the cap stage is found in the 27 mm embryo [76]. Subsequently, the tooth germ of the second deciduous molar (dm2), the epithelium thickening of the first permanent molar (M1), the presumptive primordium of the second permanent molar (M2), and the primordium of the third permanent molar (M3) are found in the 56 mm embryo, the 102 mm embryo, the 285 mm embryo, and a 4-year-old child, respectively [76]. This direction of tooth development progresses with the dynamic process that the continual lamina initiates from dm1 and extends posteriorly, interacting with the dental ectomesenchyme of a specific area and inducing the formation of deciduous and permanent molars [89].

Conversely, it remains to be discussed whether any anterior teeth in humans, including the incisors and canines, develop from the continual dental lamina (Figure 2). There is the individual development hypothesis that in humans, anterior teeth develop from isolated dental placodes and follow the first direction of tooth development, according to early observation [90]. Subsequently, Juuri and Balic [16] proposed the hybrid development hypothesis. They proposed that the deciduous lateral incisors develop from the continual lamina originating from the central incisors, and they also noted the independent development of the canines. However, no detailed evidence indicates that canines and incisors are fully independent from each other. Hence, there is another possibility that the canine is derived from the mesial lateral incisor in the continual development hypothesis. The temporal and spatial pattern of deciduous dentition development in humans, especially that of the anterior part, requires more detailed studies.

As its name implies, the continual lamina may give rise to continual teeth at the distal end of the dentition. In some patients, distal to the third molar, there is an additional molar called the distomolar or fourth molar [91]. The fourth molars resemble the third molars in shape but are much smaller in size [91]. Failed dissolution of the dental lamina may be the direct cause of fourth molars [92]. Some genes, such as *PAX9*, *SOX2*, and *WNT10A*, are likely associated with the continual dental lamina in humans, and they are closely related to tooth number abnormalities [53,93,94]. SOX2 has been reported to act as a marker in the continual lamina in humans and is expressed at the distal region of the third molar, where a fourth molar is likely to develop [93]. In summary, human posterior teeth develop from the continual lamina and play a pivotal role in tooth number regulation.

### 3.2. Mice

As a rodent with a tooth-free zone, mice can be regarded as a complicated model to study the continual lamina. As discussed above, during early mouse embryonic development, R1 and R2 (the maxillary first and second premolar rudiments, respectively) and MS and R2 (the mandibular first and second premolar rudiments, respectively) undergo cessation of growth and subsequent degeneration prior to the bud stage, resulting in the formation of the diastema [70,71]. This unique developmental pattern led to difficulties for researchers in early studies. For example, when making two-dimensional frontal sections of molar germs in murine embryos from the anterior to posterior mandible, researchers may inadvertently mix up R1 (MS), R2, and M1 because these three teeth develop subsequently in a short period in close proximity and they are similar in shape in the early stages of development, which makes it more difficult to distinguish them in continuous frontal sections of the mandible [95]. Viriot et al. [96] applied a computer-aided three-dimensional reconstruction technique to describe this process. At E12.5, MS represents the largest dental placode in the posterior region of the mandible. By E13.5, the distal end of the MS forms a wide dental bud, which briefly gives rise to R2 [96]. Subsequently, between E13.0 and E13.5, M1 arises from the distal extension of the dental lamina originating from R2. Eventually, by E14.0, R2 merges with the developing M1 [97]. The mouse molar developmental pattern is an intriguing question in developmental biology. Recently, investigations have revealed that by E14, a budding process occurs at the tail end of M1, and simultaneously, the M2 placode starts to invaginate from the epithelium, suggesting a potential dual origin for M2 [81,83]. Furthermore, M1 and M2 are connected by the continuous dental lamina at E16 [81]. Subsequently, M3 formation from the bud of the M2 distal extension has been elucidated [93,98]. This developmental pattern of a series of teeth added posteriorly is also consistent with the reiterative expression of odontogenetic genes in the region of rudiment MS, rudiment R2, and M1 during E12–14 [27]. In summary, the above studies suggest that mice undergo a continuous dental lamina developmental pattern in the diastema–molar region.

Researchers have investigated gene expression in the continual lamina in mice. *Sox2* expression is localized to the continual dental lamina [12]. *Sox2* expression is evident in the dental lamina connecting M1 and M2, as well as the bud from M2 that would develop into M3 [12]. In *Wnt10a* mutants, approximately half of the mice exhibit a fourth molar at P90 [99]. Researchers recently found that in mice with *Wnt10a* and/or *Wnt10b* mutations, there is no significant change in the number of incisors and molars in the maxilla. Moreover, they found that 60% of *Wnt10a* mutants bore a fourth molar in the mandible, a finding consistent with a previous study [73]. Moreover, elevated WNT signalling in the oral ectomesenchyme impedes the formation of M2 and M3 but does not affect M1 formation [18]. Axis inhibition protein 2 (*Axin2*), a feedback inhibitor of the WNT/beta-catenin pathway, shows high expression in the mesenchyme [18]. These findings suggest that WNT signalling may negatively regulate the progression of tooth development through the continual lamina.

Reactivating the rudimentary tooth germ in the diastema region, leading to the development of supernumerary teeth adjacent to M1, has been discussed previously [27,70,71,73,74,75]. Regression or rescue of these rudimentary tooth germs (MS, R1, and R2), resulting in additional teeth added mesial to M1, is a mechanism that influences tooth number regulation in the first direction. The second direction, namely continual tooth development, provides a potential explanation for the presence of supernumerary teeth added distal to M3 in some mutants [73] and the absence of M2 and M3 in some cases [100]. Nevertheless, there is a gap in the current research regarding detailed investigation into how the Wnt signalling pathway influences the downstream signals that impact the development of the continual lamina, particularly concerning the formation of buds from a mesial tooth such as M1. Additional comprehensive research efforts are required in this area.

### 3.3. Other Models

There is evidence of the continual lamina in other animal models, including the ferret [20], miniature pig (*Sus scrofa*) [77,78], rabbit (*Oryctolagus cuniculus*) [101,102], and Tammar wallaby (*Macropus eugenii*) [84]. In the ferret, continuous frontal sections of oral epithelium from dP4 (the fourth deciduous premolar, also the last premolar) to M1 (the first molar) confirmed the relationship between them at E34. The authors observed a dental cord connecting the M1 germ and the oral epithelium, indicating the dual origin of the M1 dental placode [103]. Interestingly, dP3 (the third deciduous premolar) is initiated much earlier than dP2 (the second deciduous premolar), indicating that the continual lamina might initiate from dP3 and extend both anteriorly and posteriorly to give rise to dP2 and dP4, respectively [103]. Similarly, dP3 is initiated first in the Tammar wallaby and produces the continual dental lamina posteriorly to generate M1 [84]. After dentition replacement, M4 is added distally to M3, increasing the tooth number by one in the permanent dentition [84].

In recent years, explanting murine tooth germs has served as a highly valuable research model [83,85]. Stabilization of WNT/beta-catenin signalling induces the formation of a series of molars [17]. Gaete et al. [83] dissected explants of the mandibular molar region and incubated them in vitro. Ablation of the M1 distal tail caused the loss of M2 tooth germ formation, and partial removal of the M1 distal tail tip preserved M2 formation, indicating the essential role of the distal lamina (continual lamina) in serial molar development. Hyaluronan (HA) expression is localized to the dental mesenchyme and in the stellate reticulum within the dental epithelium [104]. Researchers found that inhibition of HA arrested M2 formation in the explant but increased the size of M1 dramatically, as a consequence of decreased cell migration from the M1 germ tail to M2 [85]. These in vitro experiments have demonstrated that the continual lamina is a developmental pattern in the molar region and that researchers can manipulate the tooth number by regulating the formation and behavior of the continual dental lamina.

## 4. The Third Direction: Replacement/Successional Tooth Development

Animals are classified according to the number of replacement dentitions. Most mammals can change their dentition once (diphyodonts). Some mammals have only one dentition throughout their lifespan (monophyodonts), such as mice, shrews, toothed whales, etc. [86,105]. On the contrary, fish and reptiles are usually considered polyphyodonts [106]. In some species, such as *Trichiurus lepturus*, replacement teeth arise directly from the oral epithelium instead of the existing tooth germ, which has been described as the direct dental lamina [107]. In most cases, however, replacement teeth are formed from the successional (dental) lamina [12,17,108]. Different from the continual lamina in the distal end, the successional lamina extends lingually from the preceding tooth germ, while both initiate during the cap stage [12]. In general, the replacement teeth constitute the replacement dentition, and during the process of tooth replacement, the number of teeth is subject to change, which we consider to be the third direction in tooth number regulation. As continual lamina in the second direction, successional lamina is usually the core structure in the third direction (Table 1).

The tooth number remains consistent during tooth replacement. This stability is observed not only in humans—in whom five deciduous teeth are replaced sequentially by five permanent teeth within an oral quadrant—but also implied in the murine diastema, where the rudimentary tooth germ possesses a successional tooth germ of its own, termed an abortive successor [76]. A transient accessory bud on the lingual side of MS at E12.5 and M1 at E18 emerges and then rapidly disappears [97,105]. Although this successional dental lamina giving rise to the transient bud is rudimentary and soon regresses, it still has the potential to make a tooth germ [109]. The tooth number can also be altered through tooth replacement. Histological sections in miniature pigs indicate that the mandibular dP1 lacks its successor, thereby resulting in a reduction in the mandibular tooth number by one, apart from the posterior permanent molars added by the continual lamina instead of the successional lamina [78]. A similar phenomenon has been observed in the Tammar wallaby: dP2 has no successional tooth, and thus this species technically loses a premolar after tooth replacement [84]. In addition, another situation contributing to the formation of supernumerary teeth involves the successional dental lamina, when certain teeth undergo further replacement [110]. In this case, the successional dental lamina of a deciduous tooth may give rise to more than one sequentially replaced tooth on the lingual side, a phenomenon akin to what occurs in some polyphyodonts [110,111].

The successional dental lamina involves several signalling pathways. *Sox2* exhibits widespread and high expression on the lingual side of the dental stalk, from the lingual aspect of which the successional dental lamina buds and extends into the mesenchyme [12,105]. *Sox2* is required for cell proliferation during the development of the successional dental lamina [88]. WNT signalling is obviously the key regulator of successional tooth formation [16]. The expression of *Wise*, a WNT signalling modulator, is localized to the boundary between the successional dental lamina and the deciduous tooth germ [103]. WNT signalling-related genes such as *Lef1* and *Axin2* are expressed at the tip of successional dental lamina in reptiles [112]. Recently, a feedback loop has been investigated in the SOX2–WNT signalling network. *Sox2* inhibits WNT signalling while overexpression of WNT signalling downregulates *Sox2* expression in the successional dental lamina [109]. A recent study indicated that fibulin-1, secreted by fibroblasts in the dental mesenchyme, may regulate the initiation of the successional dental lamina by maintaining a niche of stem cells required for tooth germ formation [113]. SHH signalling may play an inhibitory role in the development of the successional dental lamina: In mice, interference from SHH signalling leads to further development of the rudimentary successional dental lamina of M1 [114]. These results indicate an important role for the successional dental lamina in maintaining a consistent tooth number and potentially influencing tooth number alterations.

## 5. Discussion and Perspective: Exploring the Dentition Development from the Perspective of the Three Directions

Currently, many researchers are focused on tooth number anomalies. However, there is a lack of a cohesive framework to classify tooth number regulation. For example, a report revealed that *Pax9*+/− *Eda*−/− mutant mice lack the mandibular incisors and maxillary/mandibular M3 [67]. However, the incisor and M3 may exhibit distinct growth patterns, and it is necessary to determine why they are absent. The authors discussed the hampered growth of incisors due to delayed cervical loop development [67]. On the other hand, the absence of M3 might be attributed to malformation of the continual lamina, although this potentiality has not been investigated. In addition, there have been several studies on supernumerary teeth, but some pertain to supernumerary teeth within the same dentition [115], while others refer to different dentitions [116]. Here, based on the inherent developmental patterns of teeth and dentition, we propose three directions for discussion: the development of an individual tooth, the development of a dentition (continual tooth development), and the development of multiple dentitions (successional tooth development) (Table 2).

Over the past several decades, there have been many observations and theories postulated regarding dental patterns [117]. According to Bateson’s record on dental cases, dental elements such as tooth number were variable in the distal end of dentition [117,118]. In 1938, Butler proposed the classic concept of morphogenetic fields. In Butler’s field theory, teeth do not develop as individual organs but are regulated by a systematic morphogenic field [117]. Repeated odontogenesis occurs in the odontogenic bone, and variations are distributed along the gradience of the field effect, which explains the reason for the similarity as well as the gradient difference between adjacent teeth of a dentition, especially in the posterior region [117,119]. On the basis of Butler’s field theory, Dahlberg postulated that each class of human tooth may have a field of its own [120,121]. Similarly, in Osborn’s hypothesis, there is a specific group of cells that give rise to all the teeth of the same class [122]. Our classification of tooth number alteration is complimentary to the above theories. Butler’s field effect could account for the distal extension of continual lamina, giving rise to teeth within the same dentition in the second direction of our classification. However, it is still unknown how the field exerts its effect on the continual lamina or what the underlying factor that induces the initiation of the continual lamina is.

The study of tooth number regulation connects tooth development research with clinical dental problems. Research on the tooth number provides a theoretical basis and potential therapeutic approaches for congenital tooth anomalies and contributes to tooth regeneration studies. Data from recent clinical studies also supported our classification approach. In 2021, Zhou et al. compiled the frequencies of tooth absence in humans with various gene mutations from published cases [123]. In most gene mutations, including *AXIN2*, *PAX9*, *EDA*, *EDAR*, *WNT10A*, *WNT10B*, *MSX1*, etc., there is a higher frequency of absence in the second molar than the first molar, which matches with the dynamic development process in the second dentition [123]. A similar distribution pattern of gradient tooth absence frequency in the molar region has also been reported in Fournier’s clinical review [124]. For example, in *EDA* mutant patients, the percentage of mandibular M1, M2 and M3 is about 12%, 15%, and 31%, respectively [124]. However, in their reviews, cases of deciduous dentition were not detailed. According to our classification, when reviewing cases of missing teeth, the molars in the conventional permanent dentition should be discussed and analyzed with the deciduous dentition, while the incisors, canines, and premolars in the permanent dentition should be compared with the respective precedent teeth in the deciduous dentition, which is more in line with the inherent sequence of dentition development.

Although vertebrates, including mammals, can present very different potentials for tooth replacement and dentition patterns, tooth development regulation has been conserved in vertebrates [125]. As we mentioned above, odontogenic signalling pathways such as WNT [53,73], SHH [126,127], and others have been found in different vertebrates. The polyphyodont potential has been preserved in some mammals, including humans, as a third dentition has been described in some patients [116,128]. Although mice are not the optimal choice for successional tooth studies, they remain one of the most frequently used animal models, and they can be utilized for dental-related research. For example, researchers have used mice to reveal that *Osr2* expression appears to limit the number of co-existing dentitions. An additional dentition develops on the lingual side in *Osr2* mutant mice [20,129]. Evidently, mice are good enough to investigate the first direction of tooth number regulation [130]. Due to the presence of diastema, mice also serve as a suitable model to analyze tooth development arrest [72]. Moreover, mice possess continuously growing incisors, rendering them an ideal model to study tooth regeneration. Mice can also be utilized as a model for some dental or periodontal diseases. Recently, emerging animal models such as the ferret [20], miniature pig [78], and some teleost fish [15] have been integrated into dental research. This diversification might compensate for the limitations inherent in exclusively relying on mice as the sole animal model.

The second direction, continual tooth development, is characterized by the formation of the dental lamina, giving rise to teeth within the same dentition. This process suggests that there are initiating and non-initiating teeth within one dentition, a view that corresponds with Edmund’s Zahnreihen theory that the first-formed tooth induces subsequent tooth formation [27,131,132]. Initiating teeth develop in the first direction, while non-initiating teeth are generated simultaneously through the continual lamina, which extends posteriorly, and sometimes anteriorly [20]. In the premolar and molar regions, dP3 develops first in ferret embryos, and then it gives rise to dP2 mesially and probably dP4 distally [20,103]. Interestingly, according to serial homology, the deciduous first molar in humans is an analogue of dP3 in the classic mammalian dentition pattern [95]. More evidence is required to confirm whether mammals share the same initiating posterior tooth in early odontogenesis. It remains to be studied whether it is the same story of continual development in the anterior teeth [20]. In an early report, *Wise* downregulation rescued diastema teeth in mice and added an extra pair of incisors mesial to the central incisor, suggesting that the incisor area shares a similar developmental paradigm with the diastema, where the continual lamina has been confirmed [75]. Moreover, the continual lamina is continuous with the local thickened epithelium [83]. Therefore, researchers have suggested that continual tooth development might have a dual origin—that is, from the extension of the continual tooth lamina from the proximal teeth and from placodes arising from local epithelial thickenings [76,83]. This implies an intersection between development in the first and second directions. Currently, there are some outstanding questions regarding this tooth development direction. It is unknown whether there are only two initiating teeth, namely the anterior and posterior, in one quadrant. Other topics still under investigation are what induces the initiation of the continual lamina and what feedback loop leads to its cessation in the last molar, such as M3 in humans. Furthermore, the presence of continual tooth development in other vertebrates requires further investigation.

Successional tooth development, as the third direction of development, overlaps with continual tooth development. First, they both play a role in dentition replacement. In humans, the permanent first molar is likely to derive from the continual lamina of the deciduous second molar, while the permanent first premolar comes from the successional lamina of the deciduous first molar. Second, it is noteworthy that the successional and continual laminae exhibit similar patterns in terms of gene expression. They both express *Sox2*, an odontogenic stem cell marker [12,87,133]. *Wise* is expressed lingually to the deciduous tooth germ in the successional dental lamina and buccally to the continual dental lamina [103]. Regarding the bi-allelic *WNT10A* mutation, patients are missing no more than five teeth in the deciduous dentition, while all of them present a complete congenital absence of all permanent teeth. These findings suggest that WNT10A plays a crucial role in both directions of lamina development [53]. Third, certain phenomena pose a challenge in determining the classification. For example, in the Tammar wallaby, P3 (the third permanent premolar) from the primary dental lamina replaces dP3 from the mesial side of dP3 [84]. Intriguingly, in *T. lepturus,* replacement teeth are formed directly from the oral epithelium between functional teeth [107]. In the manatee (*Trichechus inunguis*), replacement teeth are added in the present dentition distally, and these new-born molars gradually move forward to replace the worn teeth throughout life; these teeth are described as hind molars or marching molars [134,135]. These results indicate a vague boundary between replacement and continual tooth development in the oral cavity. Furthermore, although in humans dm1 gives rise to dm2 and P1 through the continual lamina and successional lamina, respectively, the contributions of dm2 or P1 to P2 remain to be investigated. Consequently, for a comprehensive investigation of successional tooth development, it is imperative to delineate the dynamic development of each tooth.

## 6. Conclusions

Over the past couple of decades, tooth number regulation has attracted intense research. This process spans from oral epithelium thickening in embryos to tooth replacement in adults. Here, we have summarized and elucidated the networks governing tooth number regulation from three distinct but closely related directions. This review has categorized the relevant studies while also highlighting a range of prospective problems. The study of tooth number regulation will enhance our understanding of tooth number anomalies in humans and lay the foundation for the application of regenerative dentistry.

## Figures and Tables

**Figure 1 ijms-24-15061-f001:**
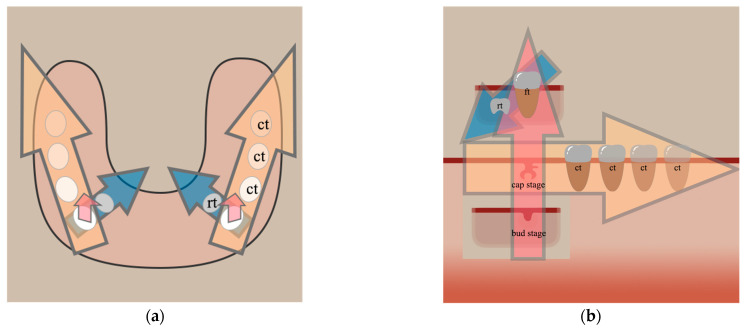
Diagram of the three directions of tooth number regulation. (**a**) A dorsal diagram of dentition development in three directions in the dental arch; (**b**) an abstract diagram of three directions of tooth number regulation. Red arrow: the first direction and its further extension refers to the process of a single tooth developing from an early stage of odontogenesis to the final functional tooth. Yellow arrow: the second direction, which is achieved by the addition of a continual tooth distal to the mesial tooth within one dentition. Blue arrow: the third direction, in which the tooth number is maintained by consistent development of a replacement tooth, usually added lingually to the functional tooth. ct: continual tooth; ft: functional tooth; rt: replacement tooth.

**Figure 2 ijms-24-15061-f002:**
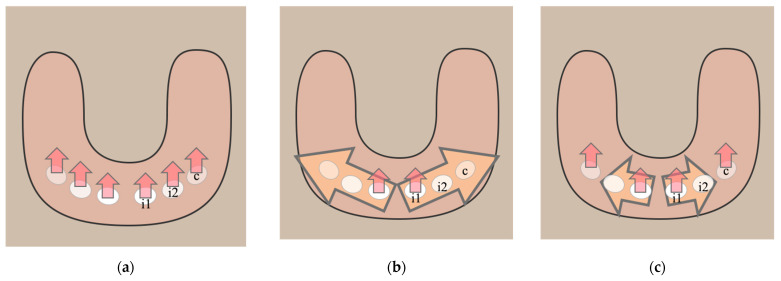
Hypotheses of anterior tooth development in human dental arch. (**a**) In the individual development hypothesis, all anterior teeth in humans develop independently of isolated dental placodes; (**b**) in the continual development hypothesis, the central incisor develops from the dental placode, and the lateral incisor and canine develop from the continual lamina from the central incisor; (**c**) in the hybrid development hypothesis, the central incisor and canine develop independently while the lateral incisor appears as part of continual development, suggesting that incisors and canines represent distinct development groups. Red arrow: the first direction of tooth number regulation. Yellow arrow: the second direction of tooth number regulation. c: canine; i1: the central incisor; i2: the lateral incisor.

**Table 1 ijms-24-15061-t001:** The usage of the term “dental lamina” in previous studies.

Type of Dental Lamina	Continual Dental Lamina	Successional Dental Lamina
Direction	The Second Direction	The Third Direction
Definition	The dental lamina extends parallel to the dentition and gives rise to teeth of same dentition.	The dental lamina gives rise to the replacement tooth lingually.
Previous descriptions of the dental lamina and relevant tooth development pattern	Additional [77,78,79] Continual [80] Continuous [81] Distal [82] Primary [12] Successional [83,84,85]	Distal [86,87] Successional [84,88]

**Table 2 ijms-24-15061-t002:** The mechanism of tooth number alteration for the three directions.

	The First Direction	The Second Direction	The Third Direction
Main role	The basis of single tooth development	Gives rise to several teeth within one dentition	Maintaining a consistent tooth number in general, and taking part in adding/reducing tooth in some cases
Mechanism of adding a tooth	The first direction itself is the process of adding a tooth	The continual lamina extends posteriorly by adding an additional tooth to the hind dentition	The successional lamina may give rise to an extra pathological replacement tooth
Mechanism of reducing a tooth	Tooth germ arrest reduces the tooth number	Arrest of continual lamina extension prevents the formation of extra teeth	Arrest of the successional lamina reduces the tooth number of the next dentition

## Data Availability

Not applicable.

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
