# Peer review of "Count Me in, Count Me out: Regulation of the Tooth Number via Three Directional Developmental Patterns"

_ijms, 2023, doi:10.3390/ijms242015061_

Round 1
Reviewer 1 Report
It has been the most pleasant and rewarding reviewing experience for me over the past year! This review manuscript is concisely written, presents clear logics and comprehensive review of the literature, and propose a novel and valuable regulatory network for understanding the tooth number regulation. I believe that this manuscript will be an important piece in the literature for the introduction of current research in tooth development. I highly recommend this review article to be published, once the authors are able to respond to the following comments.
1. Line 51, typo: “propose three directions of too number regulation”.
2. Line 80-81. “The final stage determines the morphological characteristics of the tooth and subsequent mineralization.” I am conservative on this point. The subsequent mineralization of dental hard tissues is not solely dependent upon the early tooth formation. Indeed, many of the subsequent mineralization events in tooth are quite independent of the early tooth morphogenesis. For example, many of the SCPP gene families are specialized for dentin and enamel mineralization.
3. Line 140. BMP is a family of proteins, not a single protein. Please revise accordingly.
4. Table 1 needs some major revisions. 1) The two types of dental lamina need to be further distinguished and clearly pointed out in the main text. The first category, which is mostly referred to as “continual dental lamina” in the main text, is fundamental to the “second direction”, while the second category, which is mostly referred to as “successional dental lamina” in the main text, is fundamental to the “third direction”. This has been described for the “second direction”, but not the “third direction”. It was confusing to me when I first read the session of the “third direction”, and it took me some time to refer back to this table. I suggest adding some descriptions when the authors first talk about “successional dental lamina” in the “third direction” session and refer to this table. 2) The format of this table. There should be some space or a line separating the two types of dental lamina. Also, “successional” dental lamina was repeated. In addition, I recommend highlighting the two terms that are used in this manuscript, “continual” and “successional”.
5. Line 290, typo: “that that”.
6. Figure 2 and the second paragraph of session 3.1 (Lines 294-302) need major revisions. This figure talks about 3 hypotheses, but only two of them have been explained in the main text. This paragraph discusses the “individual development hypothesis” (without mentioning this term in the main text, which should be added) and the “continual lamina hypothesis”. However, the descriptions of “continual lamina hypothesis” are conflicting with the figure. Please revise.
7. Lines 402-404. Other than mice, there are a few more monophyodont mammals. From 1) B. Berkovitz and P Shellis’s 2018 book “The teeth of Mammalian Vertebrates”, page 17, and 2) AFH van Nievelt and KK Smith’s 2005 paper “To replace or not to replace: the significance of reduced functional tooth replacement in marsupial and placental mammals.” Paleobiology 31, 324-346: “Some mammals, from unrelated groups, have ceased to replace teeth, so there is only one generation of functional teeth (monophyodonty). These include the toothed whales, the pinniped carnivorans, the shrews, the murid rodents, some moles and bats, the aardvark, and the striped skunk.” Please revise.
8. A suggestion for the authors’ consideration. The subtitle “5. Discussion and perspective” is a bit too mild and modest. One of the major contributions of this review paper is the proposal of the three directional developmental patterns. This subtitle should emphasize this concept. A potential subtitle like “The three directions of tooth number regulation based on the inherent dentition formation process” or similar might be considered.
9. Table 2. One minor issue is that the third direction, as discussed in the main text, does not always maintain a consistent tooth number in physiological development. So this description in the table seems odd to me. Please consider to revise.
Congratulations again for a wonderful review paper! I learned a lot by reviewing it. I hope to hear the authors’ responses soon.
Author Response
Reviewer 1:
Thank you! Here are the edits:
- Line 51, typo: “propose three directions of too number regulation”.
It has been corrected (Line 51).
- Line 80-81. “The final stage determines the morphological characteristics of the tooth and subsequent mineralization.” I am conservative on this point. The subsequent mineralization of dental hard tissues is not solely dependent upon the early tooth formation. Indeed, many of the subsequent mineralization events in tooth are quite independent of the early tooth morphogenesis. For example, many of the SCPP gene families are specialized for dentin and enamel mineralization.
We have rewritten this sentence (Line 79-82).
- Line 140. BMP is a family of proteins, not a single protein. Please revise accordingly.
It has been clarified BMPs are a group of proteins (Line 139-141).
- Table 1 needs some major revisions. 1) The two types of dental lamina need to be further distinguished and clearly pointed out in the main text. The first category, which is mostly referred to as “continual dental lamina” in the main text, is fundamental to the “second direction”, while the second category, which is mostly referred to as “successional dental lamina” in the main text, is fundamental to the “third direction”. This has been described for the “second direction”, but not the “third direction”. It was confusing to me when I first read the session of the “third direction”, and it took me some time to refer back to this table. I suggest adding some descriptions when the authors first talk about “successional dental lamina” in the “third direction” session and refer to this table. 2) The format of this table. There should be some space or a line separating the two types of dental lamina. Also, “successional” dental lamina was repeated. In addition, I recommend highlighting the two terms that are used in this manuscript, “continual” and “successional”.
Here are our edits according to your advice.
(1) We rewrote the paragraph introducing the third direction and highlighted the two terms by comparing the continual lamina and the successional lamina (Line 419-426).
(2) The direction of the table is flipped. It looks more intuitive. “Successional” is repeated in two directions because in those studies, some referred to “successional lamina” while some referred to “continual lamina” (Table 1, Line 289).
- Line 290, typo: “that that”.
It has been corrected (Line 298).
- Figure 2 and the second paragraph of session 3.1 (Lines 294-302) need major revisions. This figure talks about 3 hypotheses, but only two of them have been explained in the main text. This paragraph discusses the “individual development hypothesis” (without mentioning this term in the main text, which should be added) and the “continual lamina hypothesis”. However, the descriptions of “continual lamina hypothesis” are conflicting with the figure. Please revise.
We rewrote the paragraph and described the three hypotheses respectively (Line 303-311).
- Lines 402-404. Other than mice, there are a few more monophyodont mammals. From 1) B. Berkovitz and P Shellis’s 2018 book “The teeth of Mammalian Vertebrates”, page 17, and 2) AFH van Nievelt and KK Smith’s 2005 paper “To replace or not to replace: the significance of reduced functional tooth replacement in marsupial and placental mammals.” Paleobiology 31, 324-346: “Some mammals, from unrelated groups, have ceased to replace teeth, so there is only one generation of functional teeth (monophyodonty). These include the toothed whales, the pinniped carnivorans, the shrews, the murid rodents, some moles and bats, the aardvark, and the striped skunk.” Please revise.
Some other monophyodonts have been added (Line 414-416).
- A suggestion for the authors’ consideration. The subtitle “5. Discussion and perspective” is a bit too mild and modest. One of the major contributions of this review paper is the proposal of the three directional developmental patterns. This subtitle should emphasize this concept. A potential subtitle like “The three directions of tooth number regulation based on the inherent dentition formation process” or similar might be considered.
We added a subtitle ”exploring the dentition development from the perspective of the three directions” (Line 464-465)
- Table 2. One minor issue is that the third direction, as discussed in the main text, does not always maintain a consistent tooth number in physiological development. So this description in the table seems odd to me. Please consider to revise.
It has been corrected (Line 480).
Once again we sincerely appreciate your time in revising this manuscript.
Reviewer 2 Report
Dear authors,
Your manuscript presents a concise review of the information on the gene networks that control the tooth number and describes the concept of the three-dimensional regulation. Your work promotes our understanding of the mechanisms underlying odontogenesis, offers a new perspective of the successive processes and a model for categorizing the information.
Here are some suggestions and comments aiming to improve the presentation of your work.
1. Have you considered the earlier theories that have proposed models explaining the pattern of specific teeth agenesis? Your comments and considerations about the temporal and spatial pattern of the dentition development, reminded me of the models of dentition formation that have been proposed in the past. Such theories, stemming from the Butler’s field theory, had attained considerable attention in the past decades (for example, Brook, Archs Oral Biol 1984;29:373-78, Kieser, Med Hypotheses 1986;20:103-7, Townsend et al. Arch Oral Biol 2009;54:S34–44). I think it would be useful to explore whether such theories approach your opinions and can be re-examined under the light of modern discoveries and your insights on the topic.
2. I would like to see a clearer connection and/or reference to human clinical studies. Many clinical studies have appeared in the recent years presenting the phenotypes of teeth agenesis. They present the pattern of missing teeth (i.e. the most frequently missing teeth) along with genetic data of the patients. Based on such data, we expect that specific genes are mutated when certain patterns are observed. The findings of clinical publications could offer a substantial complement of your theory, and the paper would be more understandable to clinicians and clinical investigators.
3. You have presented a concise description of the role and significance of the relevant genes in the process of odontogenesis. I was somewhat surprised, though, that you had not much to say about the role of EDA and the genes of its pathway. They are among the most researched genes in connection with tooth agenesis, both syndromic and non-syndromic. I would expect that more genetic as well as clinical information is available for these genes. If it is so, this information could be useful for making your point and present a complete spectrum of the genes affecting odontogenesis and their interactions.
4. Although the figures and tables are quite illustrative, adding a graph illustrating the timeline of odontogenesis and the time periods of the gene activation would also be helpful.
Author Response
Reviewer 2.
Thank you. We have edited the text according to your suggestions.
- Have you considered the earlier theories that have proposed models explaining the pattern of specific teeth agenesis? Your comments and considerations about the temporal and spatial pattern of the dentition development, reminded me of the models of dentition formation that have been proposed in the past. Such theories, stemming from the Butler’s field theory, had attained considerable attention in the past decades (for example, Brook, Archs Oral Biol 1984;29:373-78, Kieser, Med Hypotheses 1986;20:103-7, Townsend et al. Arch Oral Biol 2009;54:S34–44). I think it would be useful to explore whether such theories approach your opinions and can be re-examined under the light of modern discoveries and your insights on the topic.
After reading reference you recommended, we find it relevant to this review, and it is added to the discussion part of the review (Line 482-497).
- I would like to see a clearer connection and/or reference to human clinical studies. Many clinical studies have appeared in the recent years presenting the phenotypes of teeth agenesis. They present the pattern of missing teeth (i.e. the most frequently missing teeth) along with genetic data of the patients. Based on such data, we expect that specific genes are mutated when certain patterns are observed. The findings of clinical publications could offer a substantial complement of your theory, and the paper would be more understandable to clinicians and clinical investigators.
A paragraph discussing the relationship between recent clinical studies and our classification is added (Line 498-515).
- You have presented a concise description of the role and significance of the relevant genes in the process of odontogenesis. I was somewhat surprised, though, that you had not much to say about the role of EDA and the genes of its pathway. They are among the most researched genes in connection with tooth agenesis, both syndromic and non-syndromic. I would expect that more genetic as well as clinical information is available for these genes. If it is so, this information could be useful for making your point and present a complete spectrum of the genes affecting odontogenesis and their interactions.
More content about EDA signaling is now included in the text (Line 209-221, 508-509).
- Although the figures and tables are quite illustrative, adding a graph illustrating the timeline of odontogenesis and the time periods of the gene activation would also be helpful.
Thank you for your precious suggestion. Actually, there was a discussion about whether putting a classic figure illustrating the process of odontogenesis when we wrote the manuscript, as it would certainly help readers to keep up with the procedures mentioned in the text. After consideration, we did not draw one. The major concern is about reinventing the wheels. There are similar illustrations (odontogenesis animation + gene expression) published in the recent years, including but not limited to:
- Bei, Marianna. 2009. “Molecular Genetics of Tooth Development.” Current Opinion in Genetics & Development, Differentation and gene regulation, 19 (5): 504–10. https://doi.org/10.1016/j.gde.2009.09.002.
- Venugopalan, Shankar R., Xiao Li, Melanie A. Amen, Sergio Florez, Diana Gutierrez, Huojun Cao, Jianbo Wang, and Brad A. Amendt. 2011. “Hierarchical Interactions of Homeodomain and Forkhead Transcription Factors in Regulating Odontogenic Gene Expression *.” Journal of Biological Chemistry 286 (24): 21372–83. https://doi.org/10.1074/jbc.M111.252031.
- Thesleff, I. 2014. “Current Understanding of the Process of Tooth Formation: Transfer from the Laboratory to the Clinic.” Australian Dental Journal 59 (s1): 48–54. https://doi.org/10.1111/adj.12102.
- Brook, Ah, J Jernvall, Rn Smith, Te Hughes, and Gc Townsend. 2014. “The Dentition: The Outcomes of Morphogenesis Leading to Variations of Tooth Number, Size and Shape.” Australian Dental Journal 59 (s1): 131–42. https://doi.org/10.1111/adj.12160.
- Matalová, Eva, Vlasta Lungová, and Paul Sharpe. 2015. “Chapter 26 - Development of Tooth and Associated Structures.” In Stem Cell Biology and Tissue Engineering in Dental Sciences, edited by Ajaykumar Vishwakarma, Paul Sharpe, Songtao Shi, and Murugan Ramalingam, 335–46. Boston: Academic Press. https://doi.org/10.1016/B978-0-12-397157-9.00030-8.
- Rostampour, Nasim, Cassy M. Appelt, Aunum Abid, and Julia C. Boughner. 2019. “Expression of New Genes in Vertebrate Tooth Development and P63 Signaling.” Developmental Dynamics 248 (8): 744–55. https://doi.org/10.1002/dvdy.26.
- Zhang, Han, Xuyan Gong, Xiaoqiao Xu, Xiaogang Wang, and Yao Sun. 2023. “Tooth Number Abnormality: From Bench to Bedside.” International Journal of Oral Science 15 (1): 5. https://doi.org/10.1038/s41368-022-00208-x.
There is a database supported by University of Helsinki (https://bite-it.helsinki.fi/).
The second concern is that list of gene expression during the process slightly drifted away from the aim of our review. Therefore, we decide not to make similar figure.
Once again we sincerely appreciate your time in revising this manuscript.